# Vitrectomy Combined with Cataract Surgery for Retinal Detachment Using a Three-Dimensional Viewing System

**DOI:** 10.3390/jcm11071788

**Published:** 2022-03-24

**Authors:** Katarzyna Nowomiejska, Mario Damiano Toro, Vincenza Bonfiglio, Aleksandra Czarnek-Chudzik, Agnieszka Brzozowska, Kamil Torres, Robert Rejdak

**Affiliations:** 1Chair and Department of General and Pediatric Ophthalmology, Medical University of Lublin, 20-079 Lublin, Poland; toro.mario@email.it (M.D.T.); olaczarnek@poczta.fm (A.C.-C.); robertrejdak@yahoo.com (R.R.); 2Eye Clinic, Public Health Department, University of Naples “Federico II”, 80131 Naples, Italy; 3Eye Clinic, Biomedicine, Neuroscience and Advance Diagnostic (BIND) Department, University of Palermo, 90143 Palermo, Italy; enzabonfiglio@gmail.com; 4Department of Mathematics and Medical Biostatistics, Medical University of Lublin, 20-090 Lublin, Poland; agnieszka.brzozowska@umlub.pl; 5Chair and Department of Didactics and Medical Simulation, Medical University of Lublin, 20-093 Lublin, Poland; kamil.torres@umlub.pl

**Keywords:** retinal detachment, vitreoretinal surgery, three-dimensional surgery system

## Abstract

Purpose: To evaluate the results of a pars plana vitrectomy (PPV) combined with cataract surgery for primary rhematogenous retinal detachment (RD) using a three-dimensional (3D) viewing system and a conventional microscope (CM). Methods: Medical reports of 82 patients were retrospectively reviewed: 26 patients were operated on with 3D and 56 patients were operated on with CM. The main outcome measures were visual acuity, duration of the surgery, and the rate of postoperative complications. Results: No statistically significant differences in pre- and postoperative visual acuity were found between both groups. There was significant improvement in the visual acuity in both groups. The best postoperative visual acuity was achieved with SF6 gas tamponade, followed by C3F8 gas and silicone oil in both groups. The duration of the surgery (60 min vs. 55 min) and the rate of postoperative complications (15% vs. 14%) were similar in both groups. Conclusions: The similar postoperative visual acuity and rate of complications detected using 3D and CM indicate that the 3D viewing system may be advantageous in the treatment of rhematogenous RD with PPV combined with cataract surgery, i.e., a complex procedure involving both anterior and posterior segment manipulations.

## 1. Introduction

Retinal detachment (RD) can lead to severe vision loss and blindness, and the only treatment restoring vision is surgery [1,2]. RD surgery is still a challenging procedure due to the heterogeneity of the disease; hence, the results of the surgical treatment vary between centers and surgeons. Pars plana vitrectomy (PPV) is now considered a primary strategy for RD repair by many surgeons [3,4] with a current anatomical success rate of up to 80% [5,6]. There is a wide range of indications in RD, as there may be different rhematogenous, traumatic, and tractional origins due to diabetic retinopathy or accompanying high myopia. Currently, the advantage of RD surgery consists of combining PPV with cataract surgery. It has already been found to be an effective and safe procedure to treat RD [7,8] with a conventional microscope (CM).

However, visualization during RD surgery may dramatically limit the anatomical and functional outcome. Non-contact wide-field viewing systems introduced by Landers in 2003 led to considerable improvement in the field [9]; however, it has many limitations. Moreover, the position of the surgeon’s body may be uncomfortable during longer surgeries.

The introduction of the three-dimensional (3D) digitally assisted visualization system is an enormous advance in ophthalmic surgery in recent years [10]. The prototype for 3D was first introduced in 1995 as a laboratory microsurgery system carried out using a 3D screen. The 3D technology is based on high-resolution dual-camera systems retransmitting an image on a 55-inch screen in front of the surgeon, the rest of the surgical team, students, and ophthalmologists in training [11]. In the current “heads-up” display systems, two images are mixed horizontally and then separated using polarized glasses so that each eye sees a slightly horizontally disparate image, which allows perception of depth [10].

The 3D system offers many advantages, e.g., a comfortable position for the surgeon (“heads up”), a higher resolution image, a lower endoillumination level [12,13], and higher magnification than in the traditional viewing system [14]. The 3D technology has been improved for many years, and many commercially available 3D systems for eye surgeries are now available in clinical practice.

The purpose of this study was to analyze the success rate of surgical management of RD with vitrectomy combined with cataract surgery performed at a large university hospital with different viewing systems: CM and the 3D imaging system.

## 2. Materials and Methods

The study was planned as a monocentric, retrospective, observational, case-controlled study over a two-year period. Approval of the Ethics Committee of the Medical University of Lublin, Poland, was obtained (approval number KE-0254/255/2021). The study was performed in accordance with the tenets of the Declaration of Helsinki. Two groups of consecutive patients were compared: one group was operated on with CM and the second group was operated on with the 3D imaging system as part of routine care. The inclusion criteria were as follows: above 50 years of age, PPV was performed due to primary rhematogenous RD, lens opacity was exceeding the first grade of each category of the Lens Opacities Classification System III (LOCS III) scale. In turn, previous cataract surgery and vitrectomy were the exclusion criteria. Moreover, trauma, high myopia (>6D), and diabetic retinopathy were exclusion criteria. All consecutive cases operated on in 2019 and 2020 were included. Overall, 26 patients were operated on with the 3D system and 56 patients were operated on using CM. The mean age of the patients in both groups was 58 (range 51–68) years. There were no significant differences between the groups in regard to age (*p* = 0.83). The 3D group consisted of 62% males and 38% females. The proportion of males to females in the CM group was 55% to 45%.

All patients underwent a detailed ophthalmologic examination before and after surgery, including the determination of best-corrected visual acuity (BCVA) measured with projected-light Snellen charts. The results of the Snellen visual acuity were converted to the logMAR (minimum angle of resolution) scale. Subjects with counting fingers, hand motion, light perception, or no light perception visual acuity were assigned with logMAR values of 1.9, 2.3, 2.7, and 3.0, respectively [15]. Moreover, biomicroscopy along with anterior segment evaluation, fundus examination, and careful peripheral retinal examination were performed. The axial length (AL) was measured preoperatively using optical biometry (IOL Master; Carl Zeiss Meditec AG, Jena, Germany) and checked with A-scan ultrasound (OcuScan RxP Ophthalmic Ultrasound System; Alcon Laboratories, Fort Worth, TX, USA). The IOL power calculation was performed using the SRK/T formula.

An OPMI Lumera 700 microscope (Carl Zeiss Meditec AG, Jena, Germany) was used as a CM, in addition to a non-contact wide-angle BIOM Viewing System. The Ngenuity 3D Visualization System (Alcon Laboratories, Fort Worth, TX, USA) v1.2.9 software version was used for visualization of the surgery in the 3D group of patients. The high-definition real-time video was displayed on a flat-panel placed at 1.3 m from the surgeon (Appendix A).

All cases were performed using a standard technique by two experienced vitreoretinal surgeons (RR and KN) with a local (peribulbar) anesthesia. The surgical procedure based on the 3D technique was no different from the surgery performed using CM. All eyes underwent extensive 3-port PPV using 23 or 25 gauge instrumentation (Constellation, Alcon, Fort Worth, US) followed by fluid–air exchange, endophotocoagulation around the break(s), and gas (25%) SF6 or (12%) C3F8 or 5000 cSt silicone oil as a tamponade according to the nature of the RD.

Phacoemulsification was performed before the PPV through a 2.8 mm clear corneal incision, with implantation of an acrylic foldable intraocular lens (IOL) AcrySof MA50BM (Alcon Laboratories, Fort Worth, TX, USA). Central and complete peripheral vitrectomy as well as posterior hyaloid detachment was performed in all cases without triamcinolone staining. Brilliant blue G (DORC, Zuidland, the Netherlands) or indocyanin green was used to stain the internal limiting membrane (ILM) in cases with “macula off” RD (15 cases in the 3D group, or 58% and 32 cases the CM group, or 57%).

Details on the surgery performed, preoperative characteristics, and follow-up were collected from the hospital database after anonymization. All patients received a topical anti-inflammatory drug (steroid) for 4 weeks after surgery and an antibiotic for 1 week. The preoperative data included the patients’ age and gender, AL, best-corrected visual acuity (BCVA) measured with Snellen charts, and characteristics of the RD. The intraoperative data included the type of tamponade and the complications encountered during the surgery. The postoperative data included the primary and final anatomical success rate, BCVA, and any complications. The main outcomes were the rate of retinal reattachment, best-corrected visual acuity (BCVA) using Snellen charts, and the duration of the surgery.

### Statistical Analysis

Statistical computations were performed with STATISTICA 13.0 computer software (StatSoft, Kraków, Poland). The values of the analyzed measurable parameters were presented as the mean and standard deviations (SD) and as counts and percentages in the case of non-measurable parameters. The Mann–Whitney test was used to compare independent groups. The non-parametric statistical Wilcoxon signed-rank test was used to compare dependent variables of two related samples. The normal distribution of the measurable parameters was assessed using the Shapiro–Wilk test. Spearman’s rank correlation coefficient was used to assess the relationship between the variables. A level of significance of *p* < 0.05 indicated the existence of statistically significant differences.

## 3. Results

The median preoperative visual acuity was 1.3 logMAR in the 3D group and 2.0 logMAR in the CM group and there were no significant differences between the groups (*p* = 0.88). The median postoperative visual acuity was also no different than the 3D group and the CM group and was 0.10 and 0.15, respectively (*p* = 0.06). There was significant improvement in the visual acuity in both groups after the operation; in the 3D group (Z = 2.63; *p* = 0.009) and in the CM group (Z = 5.12; *p* < 0.000001).

Considering the tamponade used at the end of the surgery, there were significant differences in both groups between eyes treated with silicone oil, SF6 gas, and C3F8 tamponade in regard to postoperative visual acuity (Table 1 and Table 2).

The median duration of the surgery was 60 min (range: 55–70 min) in the 3D group and 55 min (range: 50–65 min) in the CM group (*p* = 0.13). The patients reported no intolerable discomfort or glare during the surgery. The median follow-up period was 160 days (77–279 days) in the 3D group and 82 days (30–219 days) in the CM group (*p* = 0.02).

Regarding complications, there were two eyes with secondary RD, one eye with cystoid macular edema, and one eye with secondary glaucoma in the group operated on with 3D. In the group operated on with CM, there were no eyes with secondary RD, one eye with hypotony, five eyes with secondary glaucoma, and eyes with epiretinal membrane. Overall, the rate of the postoperative complications was similar (*p* = 0.08) in both groups (15% vs. 14%) (Table 3).

## 4. Discussion

To the best of our knowledge, this is the first study describing results consisting only of vitreoretinal surgery combined with cataract surgery in the 3D viewing system. The 3D display has already been used during cataract surgery alone [16], PPV alone [17,18,19], or PPV combined with cataract surgery in a minority of cases [20,21]. PPV combined with cataract surgery using the “heads-up” system has already been described as part of the case series. For example, Matsumoto described the results of 3D surgery where 5 of the 74 eyes underwent phacovitrectomy [13].

In our series of 82 patients, the rate of complications at 6 months after PPV with 3D was 15%; in the CM group, the rate was 14%. The 3D group showed a tendency toward a higher incidence of retinal redetachment. We can anticipate that it may be due to the “learning effect”, as the number of the 3D surgeries is lower than the number of surgeries performed with CM. The data reported from prior studies investigating this issue showed that the rate of secondary RD was similar to that in the CM group [22] or higher in the 3D group [20]. In a study conducted by Zhang, the rate of complications was substantially higher but similar in both 3D (30.6%) and TM groups (30.2%) [23]. However, Zhang and colleagues analyzed very difficult cases of RD complicated by proliferative vitreoretinopathy, recurrent RD and tractional RD due to diabetic retinopathy. In our study, all eyes were operated on due to primary rhematogenous RD.

Phacoemulsification combined with PPV is a complex and challenging surgery and requires not only the advanced surgical skills of the surgeon but also good visualization both of the anterior and posterior segments of the eye. This surgery has many advantages, e.g., enhanced retinal visualization during posterior segment surgery, better access to the vitreous base allowing more extensive vitrectomy and endolaser treatment, more extensive gas filling, and better tamponade of retinal breaks. However, there is a risk of postoperative refractive errors, especially in macula-off cases, iatrogenic anisometropia in myopic subjects, and removal of the lens in cases of no cataract with residual accommodative function [7]. Moreover, there are other disadvantages, such as a longer operation time, loss of corneal transparency during the procedure, and risks of posterior or IOL dislocation or capture [19]. Vitrectomized eyes are also at a higher risk of developing intraoperative complications, such as posterior capsular rupture, zonular dialysis, and dislocated lens nuclear fragments. Based on the results of our study we can conclude that the “heads up” viewing system is as good as CM in performing PPV combined with cataract surgery due to rhematogenous RD, as final visual acuity and rate of complications are similar in both viewing systems.

Vitreoretinal surgery using the 3D viewing system has already been described [24], mostly for macular hole surgery [17,22,25]. All the authors have shown that both anatomical and functional results of surgery in the 3D viewing system are not worse than those of CM surgery [10,24,26]. The disadvantages of the 3D system include the potential longer surgical time, the steep learning curve, and uncomfortable visual perception [14]. In our study, there were no significant differences in the operation duration (60 vs. 55 min); however, in a majority of studies, 3D surgery took significantly longer [22,23,24].

The “heads-up” display is superior to CM surgery in many aspects. The 3D equipment can provide extended field depth (5 times better than CM) due to the better light sensitivity of the software and cameras [12]; moreover, the surgeon needs less accommodative effort [27] and lower endoillumination levels are needed, without loss of the quality of the view [12,28,29], thus retinal phototoxicity is reduced, which is important during surgical manipulations in the vitreous cavity. The magnification of the image is 48% higher than in CM and there is a 42% increase in depth resolution to resolve fine details [12]. The 3D system is also preferred by surgeons [28] due to the educational possibilities [24], larger field of view, better resolution, better ergonomics, and lesser neck and back pain [21]. In a recent meta-analysis published in 2022 by Wang [30] analyzing 15 studies and 2889 eyes operated on with the ‘heads-up’ display, this technique was found equal to CM, as vitrectomy with 3D took a longer surgical time but needed lower endoillumination levels. There were no significant differences in the postoperative best-corrected visual acuity, intraoperative complications, and reattachment rate of RD. It has already been reported that 3D visualization allows avoiding the use of triamcinolone to stain the vitreous area [18]. This may be related to the fact that vitreous remnants are very well visible in digital (green) filters and the magnified high-resolution image during 3D surgery. We used no triamcinolone in our case series in either the 3D or CM group.

Our study is limited by its retrospective nature and the smaller number of cases in the 3D group vs. the CM group and the fact that two surgeons performed the surgeries. However, only cases of pure rhematogenous RD have been included and the follow-up period is quite long (median 160 days in the 3D group). Another strength of the study is that there is a huge similarity between the groups in regard to use of tamponades.

## 5. Conclusions

In conclusion, the “heads-up” viewing system is not inferior to CM in patients with rhematogenous RD treated with PPV combined with cataract surgery, which is a more demanding surgery as it requires both anterior and posterior segment manipulations involving ILM peeling. In the present study, no significant differences were found in regard to the duration of surgery, rate of complications, and functional results.

## Figures and Tables

**Table 1 jcm-11-01788-t001:** Values of the postoperative visual acuity (mean, standard deviation, median, lower and upper quartile) in logMAR in eyes treated with different tamponades (silicone oil, SF6 gas, and C3F8 gas) in the group operated on with the 3D imaging system.

Tamponade	Mean Visual Acuity	Standard Deviation	Lower Quartile	Median	Upper Quartile
Silicone oil	1.29	0.71	0.70	1.00	2.00
C3F8 gas	0.89	0.43	0.52	1.00	1.30
SF6 gas	0.52	0.35	0.22	0.52	0.70
H = 6.12, *p* = 0.05 *					

* The results of statistical analysis.

**Table 2 jcm-11-01788-t002:** Values of the visual acuity (mean, standard deviation, median, lower and upper quartile) in logMAR in eyes treated with different tamponades (silicone oil, SF6 gas, and C3F8 gas) in the group operated on with the conventional microscope.

Tamponade	Mean Visual Acuity	Standard Deviation	Lower Quartile	Median	Upper Quartile
Silicone oil	1.07	0.53	0.70	1.00	1.30
C3F8 gas	0.80	0.37	0.52	0.85	1.00
SF6 gas	0.44	0.31	0.15	0.40	0.70
H = 20.36, *p* < 0.0001 *					

* The results of statistical analysis.

**Table 3 jcm-11-01788-t003:** Rate of postoperative complications in the three-dimensional viewing system (3D) and conventional microscope (CM) groups expressed in percent (%) and number of eyes (*n*).

Patient Group	Non-Retinal Detachment Other Complications			Overall *n* (%)
	*n* (%)	*n* (%)	*n* (%)	
3D	22	2	2	26
	84.62%	7.69%	7.69%	100.00%
CM	48	0	8	56
	85.71%	0.00%	14.29%	100.00%
Overall	70	2	10	82
	85.37%	2.44%	12.20%	100.00%
Chi^2^ = 4.94; *p* = 0.08				

## Data Availability

Data are available on reasonable request to the corresponding author.

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
