# Peer review of "Vitrectomy Combined with Cataract Surgery for Retinal Detachment Using a Three-Dimensional Viewing System"

_jcm, 2022, doi:10.3390/jcm11071788_

Round 1

Reviewer 1 Report

Comments:

The authors used a new type of three-dimensional viewing system to facilitate vitrectomy and cataract surgery in retinal detachment. Compared with conventional microscope, the novel viewing system seems to be the same effective. However, there are some defects in study design and analysis.

  1. In this manuscript, the authors choose best-corrected visual acuity (BCVA) measured with projected-light Snellen charts as main outcome. It is ok to choose BCVA as a postoperative prognostic indicator. However, Snellen charts could not be used directly in statistic analysis. Please convert the result of Snellen charts into LogMar.
  2. Postoperative cataract is commonly observed in eyes with PPV history. However, PPV combined with cataract surgery is not a conventional surgery strategy in retinal detachment eye, especially in those with silicon oil tamponade. However, the authors said nothing about the operation indication of combined surgery. Please explain why you choose combined surgery instead of PPV with subsequent cataract surgery.
  3. The authors include many different types of retinal detachment to observe. The most type is rhematogenous RD. However, the postoperative BCVA varies dramatically in different causes of RD. Please exclude other causes of RD including traumatic, diabetic and high myopia RD and include only one cause of RD.

Author Response

Dear Reviewer,

Thank You very much for the revision and a number of constructive suggestions of the manuscript "Vitrectomy combined with cataract surgery for retinal detachment using a three-dimensional viewing system".

We are very grateful for the opportunity to submit the revised manuscript and Your help in improving it. We strictly followed the Reviewer's suggestions, as is presented below and in the manuscript with track changes.

The authors used a new type of three-dimensional viewing system to facilitate vitrectomy and cataract surgery in retinal detachment. Compared with conventional microscope, the novel viewing system seems to be the same effective. However, there are some defects in study design and analysis.

In this manuscript, the authors choose best-corrected visual acuity (BCVA) measured with projected-light Snellen charts as main outcome. It is ok to choose BCVA as a postoperative prognostic indicator. However, Snellen charts could not be used directly in statistic analysis. Please convert the result of Snellen charts into LogMar.

In the methods section it has been added (line 95-98):

The results of the visual acuity were converted into logMAR (minimum angle of resolution) scale.  Subjects with counting fingers, hand motion, light perception, or no light perception visual acuity were assigned with values of logMAR of 1.9, 2.3, 2.7, and 3.0, respectively [13].

In the results section it is now written (line 159-179):

The median preoperative visual acuity was 1.3 logMAR in the 3D group and 2.0 logMAR in the CM group and did not differ significantly between the groups (p=0.88). The median postoperative visual acuity also did not differ between the 3D group and the CM group and was 0.10 and 0.15, respectively (p=0.06). There was significant improvement in the visual acuity in both groups after the operation in the 3D group (Z=2,63; p=0,009) and in the CM group (Z=5,12; p<0,000001).

Table 1 and 2 have been changed as follows (lines 183-190):

Table 1. Values of the postoperative visual acuity (mean, standard deviation, median, lower and upper quartile) in logMAR in eyes treated with different tamponades (silicone oil, SF6 gas, and C3F8 gas) in the group operated with the 3D imaging system.

Tamponade

Mean

Visual acuity

Standard deviation

Lower quartile

Median

Upper quartile

Silicone oil        

1.29

0.71

0.70

1.00

2.00

C3F8 gas

0.89

0.43

0.52

1.00

1.30

SF6 gas

0.52

0.35

0.22

0.52

0.70

H=6.12, p=0.05*

                          *the results of statistical analysis

Table 2. Values of the visual acuity (mean, standard deviation, median, lower and upper quartile) in logMAR in eyes treated with different tamponades (silicone oil, SF6 gas, and C3F8 gas) in the group operated with the conventional microscope.

Tamponade

Mean

Visual acuity

Standard deviation

Lower quartile

Median

Upper quartile

Silicone oil

1.07

0.53

0.70

1.00

1.30

C3F8 gas

0.80

0.37

0.52

0.85

1.00

SF6 gas

0.44

0.31

0.15

0.40

0.70

H=20.36, p<0.0001*

                          *the results of statistical analysis

Instead of the following reference:

  1. Holladay, J.T. Proper method for calculating average visual acuity. J Refract Surg. 1997, 13, 388–391.

Another one has been inserted in the references section:

  1. Schulze-Bonsel K., Feltgen N., Burau H., Hansen L., Bach M. Visual acuities “hand motion” and “counting fingers” can be quantified with the freiburg visual acuity test. Investig. Ophthalmol. Vis. Sci. 2006;47:1236–1240.

Postoperative cataract is commonly observed in eyes with PPV history. However, PPV combined with cataract surgery is not a conventional surgery strategy in retinal detachment eye, especially in those with silicon oil tamponade. However, the authors said nothing about the operation indication of combined surgery. Please explain why you choose combined surgery instead of PPV with subsequent cataract surgery.

Only cases with cataract have been included.

In the methods section it is now written (line 82-85):

The inclusion criteria were as follows: age above 50 years, PPV performed due to primary rhematogenous RD, lens opacity exceeding the first grade of each category of the Lens Opacities Classification System III (LOCS III) scale.

The authors include many different types of retinal detachment to observe. The most type is rhematogenous RD. However, the postoperative BCVA varies dramatically in different causes of RD. Please exclude other causes of RD including traumatic, diabetic and high myopia RD and include only one cause of RD.

Only pure primary rhematogenous retinal detachment cases have been included. Eyes with RD and trauma, diabetic retinopathy and high myopia have been excluded.

The following sentence has been excluded from the results section (line 21):

Both groups included eyes with rhematogenous RD or RD associated with myopia, diabetic retinopathy or trauma.

The following sentences have been excluded from the results section (line 160):

The preoperative diagnosis did not differ significantly between the groups (p=0.48). The majority comprised eyes with rhematogenous RD (83% in the 3D group and 89% in the CM group); the minority were cases of RD of another origin – traumatic and tractional due to diabetic retinopathy and high myopia (17% and 11% respectively).

In the abstract it is now written (line 18-21):

Purpose: To evaluate the results of pars plana vitrectomy (PPV) combined with cataract surgery for primary rhematogenous retinal detachment (RD) using a three-dimensional (3D) viewing system and a conventional microscope (CM). Methods: Medical reports of 82 patients were retrospectively re-viewed:  26 patients operated with 3D and 56 patients operated with CM.

And later (line 26-27): The duration of the surgery (60 min vs. 55 min) and the rate of postoperative complications (15% vs. 14%) were similar in both groups.

In the methods section it has been corrected as follows (line 86-87):

In turn, previous cataract surgery and vitrectomy were the exclusion criteria. Moreover trauma, high myopia (>6D) and diabetic retinopathy were exclusion criteria.

 And later (line 88-92): Overall, 26 patients were operated with the 3D system and 56 patients were operated using CM. The mean age of the patients in both groups was 58 (rage 52-68) years. There were no significant differences between the groups in regard to age (p=0.83). In the 3D group, there were 62% of males and 38% of females. The proportion of males to females in the CM group was 55% to 45%.

In the results section it is now written (line 199-422):

As regards complications, there were two eyes with secondary RD, 1 eye with cystoid macular edema, and 1 eye with secondary glaucoma in the group operated with 3D. In the group operated with CM, there were no eyes with secondary RD, 1 eye with hypotony, 5 eyes with secondary glaucoma, and eyes with epiretinal membrane. Overall, the rate of the complications was similar (p=0.08) in both groups (15% vs. 14%) (table 3).

Table 3. Rate of postoperative complications in the 3 dimensional viewing system (3D) and conventional microscope (CM) groups expressed in percent (%) and number of eyes (n).

Patient group

None    Retinal detachment  Other complications

 Overall

n (%)

n (%)

  n (%)  

n (%)

3 D

22

2

2

26

84.62%

7.69%

7.69%

100.00%

CM

48

0

8

       56

85.71%

0.00%

14.29%

100.00%

Overall

70

2

10

82

85.37%

2.44%

12.20%

100.00%

Chi2=4.94; p=0.08

                          *the results of statistical analysis

In the discussion it is now written (line 440-441):

In our study all eyes were operated due to primary rhematogenous RD.

And later (line 595-596):

However, only cases of pure rhematogenous RD have been included and the follow-up period is quite long (median 160 days in the 3D group).

Reviewer 2 Report

Since the number of heads-up vitrectomy has been growing, this study will also support the usefulness of this technique. The interesting part of this study is to focus on the subjects with RD patients. However, there are some concerns in the context of this work toward publication.

Major points

This study included not only rhegmatogenous retinal detachment but also tractional detachment in diabetic retinopathy. Because there is a huge difference in the pathogenesis of these situation, the severity of diabetic retinopathy might affect the results of this study. Please describe the severity of diabetic retinopathy. And was pan photocoagulation done before the surgery?

Minor points

  • Line 108. The authors mentioned about the use of Brilliant blue G or ICG. But they did not describe how often they used in both groups and what cases they performed ILM peeling.

  • To make readers to easily understand the difference in the groups, Table should be prepared for the characteristics of these groups instead of writing in the results section.

Author Response

Dear Reviewer,

Thank You very much for the revision and a number of constructive suggestions of the manuscript "Vitrectomy combined with cataract surgery for retinal detachment using a three-dimensional viewing system".

We are very grateful for the opportunity to submit the revised manuscript and Your help in improving it. We strictly followed the Reviewer's suggestions, as is presented below and in the manuscript with track changes.

Since the number of heads-up vitrectomy has been growing, this study will also support the usefulness of this technique. The interesting part of this study is to focus on the subjects with RD patients. However, there are some concerns in the context of this work toward publication.

Major points

This study included not only rhegmatogenous retinal detachment but also tractional detachment in diabetic retinopathy. Because there is a huge difference in the pathogenesis of these situation, the severity of diabetic retinopathy might affect the results of this study. Please describe the severity of diabetic retinopathy. And was pan photocoagulation done before the surgery?

As suggested also by the second reviewer, only pure primary rhematogenous retinal detachment cases have been included. Eyes with RD and trauma, diabetic retinopathy and high myopia have been excluded.

The following sentence has been excluded from the results section (line 21):

Both groups included eyes with rhematogenous RD or RD associated with myopia, diabetic retinopathy or trauma.

The following sentences have been excluded from the results section (line 160):

The preoperative diagnosis did not differ significantly between the groups (p=0.48). The majority comprised eyes with rhematogenous RD (83% in the 3D group and 89% in the CM group); the minority were cases of RD of another origin – traumatic and tractional due to diabetic retinopathy and high myopia (17% and 11% respectively).

In the abstract it is now written (line 18-21):

Purpose: To evaluate the results of pars plana vitrectomy (PPV) combined with cataract surgery for primary rhematogenous retinal detachment (RD) using a three-dimensional (3D) viewing system and a conventional microscope (CM). Methods: Medical reports of 82 patients were retrospectively re-viewed:  26 patients operated with 3D and 56 patients operated with CM.

And later (line 26-27): The duration of the surgery (60 min vs. 55 min) and the rate of postoperative complications (15% vs. 14%) were similar in both groups.

In the methods section it has been corrected as follows (line 86-87):

In turn, previous cataract surgery and vitrectomy were the exclusion criteria. Moreover trauma, high myopia (>6D) and diabetic retinopathy were exclusion criteria.

 And later (line 88-92): Overall, 26 patients were operated with the 3D system and 56 patients were operated using CM. The mean age of the patients in both groups was 58 (rage 52-68) years. There were no significant differences between the groups in regard to age (p=0.83). In the 3D group, there were 62% of males and 38% of females. The proportion of males to females in the CM group was 55% to 45%.

In the results section it is now written (line 199-422):

As regards complications, there were two eyes with secondary RD, 1 eye with cystoid macular edema, and 1 eye with secondary glaucoma in the group operated with 3D. In the group operated with CM, there were no eyes with secondary RD, 1 eye with hypotony, 5 eyes with secondary glaucoma, and eyes with epiretinal membrane. Overall, the rate of the complications was similar (p=0.08) in both groups (15% vs. 14%) (table 3).

Table 3. Rate of postoperative complications in the 3 dimensional viewing system (3D) and conventional microscope (CM) groups expressed in percent (%) and number of eyes (n).

Patient group

None    Retinal detachment  Other complications

 Overall

n (%)

n (%)

  n (%)  

n (%)

3 D

22

2

2

26

84.62%

7.69%

7.69%

100.00%

CM

48

0

8

       56

85.71%

0.00%

14.29%

100.00%

Overall

70

2

10

82

85.37%

2.44%

12.20%

100.00%

Chi2=4.94; p=0.08

                          *the results of statistical analysis

In the discussion it is now written (line 440-441):

In our study all eyes were operated due to primary rhematogenous RD.

And later (line 595-596):

However, only cases of pure rhematogenous RD have been included and the follow-up period is quite long (median 160 days in the 3D group).

Minor points

Line 108. The authors mentioned about the use of Brilliant blue G or ICG. But they did not describe how often they used in both groups and what cases they performed ILM peeling.

In the methods section the following phrase has been added (line 136-138):

Brilliant blue G (DORC, Zuidland, the Netherlands) or indocyanin green was used to stain the internal limiting membrane (ILM) in cases with “macula off” RD (15 cases in 3D group -58% and 32 cases in CM group-57%).

To make readers to easily understand the difference in the groups, Table should be prepared for the characteristics of these groups instead of writing in the results section.

As only one group was left (primary rhematogenous retinal detachment) we have not created any additional table comparing groups with RD of different origin.

Round 2

Reviewer 1 Report

The manuscript were revised accordingly and no any comments to authors.